# Tracking the Evolution of Microbial Communities on Microplastics through a Wastewater Treatment Process: Insight into the "Plastisphere"



**Jin-Kyung Hong** [1], **Hyecheol Oh** [2], **Tae Kwon Lee** [1], **Seogku Kim** [2,3], **Daemin Oh** [2], **Jaehwan Ahn** [2] and **Saerom Park** [2,3,*]

1  Department of Environmental and Energy Engineering, Yonsei University, Wonju 26493, Republic of Korea; hongjk@yonsei.ac.kr (J.-K.H.); tklee@yonsei.ac.kr (T.K.L.)
2  Department of Environmental Research, Korea Institute of Civil Engineering and Building Technology (KICT), Goyang-si 10223, Republic of Korea; hcoh79@kict.re.kr (H.O.); sgkim@kict.re.kr (S.K.); ohdm78@naver.com (D.O.); jhahn@kict.re.kr (J.A.)
3  Department of Civil and Environmental Engineering, University of Science and Technology, Daejeon 34113, Republic of Korea
*  Correspondence: srpark@kict.re.kr; Tel.: +82-31-910-0628

**Abstract:** Microplastics (MPs), which result from the breakdown of plastic waste, have become ubiquitous in various environmental compartments. The "plastisphere", referring to the unique bacterial communities inhabiting plastic debris, includes pathogens and antibiotic resistance genes. Wastewater treatment plants (WWTPs) are hotspots for plastisphere formation, but significant releases of MPs still occur. This study investigates the microbial communities on polystyrene (PS) MPs through in situ deployment across primary, secondary, and tertiary WWTP stages. Biofilms formed on the PS MPs exhibited greater bacterial diversity than background waters. Certain genera acted as pioneers in the biofilms, attracting and facilitating the accumulation of other microbes from background waters. The biofilms formed on the MPs became more resistant to treatment processes compared to freely floating bacteria. This study sheds light on the evolution of microbial communities on MPs within WWTPs and their roles as carriers of microbes in effluents, with implications for environmental and public health. Understanding these dynamics is crucial for effective control over MPs and microbial pollution in WWTPs.

**Keywords:** microplastics; polystyrene; microbial community; plastisphere; wastewater treatments





## 1. Introduction

Plastics have gained immense popularity as synthetic materials due to their high durability, lightweight nature, and cost-effectiveness, resulting in the production of millions of tons every year [1,2]. However, only a small proportion of plastics are recycled or incinerated, while the majority accumulate as plastic waste in the environment. Microplastics (MPs), defined by the NOAA as plastic fragments ranging in size from 1 μm to 5 mm, are generated from larger plastic items through processes such as biological, chemical, and physical weathering [1,3,4]. As a consequence, MPs are pervasive across various environmental domains such as soils, aquatic ecosystems, wildlife habitats, and even the atmosphere [5–8].

The term "plastisphere" refers to the unique bacterial communities that inhabit plastic debris, characterized by interactions between co-occurring organisms in their environment [9,10]. Wastewater treatment plants (WWTPs) have been recognized as hotspots for the formation of plastispheres, functioning as transporters of bacteria [11,12]. Despite achieving high removal rates of 90~99% for MPs through WWTP processes [13–16], a significant fraction of MP particles ranging from $2 \times 10^6$~$4 \times 10^9$ MPs per day can still be released

from a single WWTP effluent, raising concerns regarding the transport of MPs carrying bacteria [17–19]. Notably, Kelly et al. (2021) [20] reported high abundances of *Xanthomonas* and *Campylobacteraceae* on fiber types of MPs, while *Betaproteobacteria* and *Gammaproteobacteria* were found to be prevalent on the surface of PE in an effluent [21]. Other recent studies demonstrated how MPs can serve as protective habitats for pathogenic bacteria or ARGs such as *Streptococcus*, *Pseudomonas*, *Lactobacillus*, and *Acinetobacter*, enabling them to survive disinfection processes referred to as the "umbrella effect" [21–24].

However, the existing studies primarily focused on analyzing the microbial community compositions of biofilms formed on MPs under controlled conditions that often represented fragmented segments of the WWTP process. These conditions often replicated specific treatment stages within the WWTP process, but they often did not provide insights into the consecutive formation of biofilms on MPs as they progressed through the entire WWPT treatment process. Moreover, these studies mainly centered on target microorganisms such as human microbiota or pathogens, neglecting the broader microbial diversity present on MPs [20–24]. Considering that the formation and maturation of biofilms are significantly influenced by environmental conditions [10,25], it is crucial to investigate the dynamic changes and distinctive features of the microbial communities that develop on MPs via in situ incubation within the real WWTP process.

Hence, this study aimed to investigate variations in the characteristics of microbial communities formed on the surfaces of MPs as they traversed the primary, secondary, and tertiary treatment stages of WWTPs. We hypothesized that the compositions of the microbial communities on MPs are altered as they pass through each WWTP stage. To gain insight into the influences of the abiotic and biotic factors present in WWTPs, as well as the impact of previous stages on the formation of biofilms on MPs in a real-world setting, all experiments were conducted via in situ incubation within the operational stages, ranging from the primary to tertiary treatment of WWTPs. Polystyrene (PS) was selected as a representative MP due to its high detection frequency in WWTPs [26]. Granular pieces of PS, with sizes ranging from 2 to 3 mm, were sequentially deployed from the primary to the tertiary treatment stages for a duration of 7 days per treatment. The physical morphology of the biofilm was analyzed using microscopic techniques, while the compositions of microbial communities within the biofilms on the MPs were investigated using a next generation sequencing (NGS) analysis. By examining the microbial communities on MPs across the WWTP stages, this study will enhance our understanding of how these communities evolve and adapt within the complex WWTP environment. Furthermore, the findings will provide valuable insights into the potential role of MPs as microbial carriers and contribute to the development of effective strategies for MPs and associated microbial pollution control in WWTPs.

## 2. Materials and Methods

### 2.1. Description of the WWTP

Samplers containing MPs were deployed from 21 February to 14 March 2022 at a WWTP facility located in Ilsanseo-Gu, Goyang-Si, Gyeonggi-Do, South Korea ($37°39'33''$ N, $126°43'30''$ E). The WWTP in Ilsanseo-Gu consists of a tertiary advanced-treatment system with a capacity of 270,000 $m^3$/day, an average inflow rate of 189,265 $m^3$/day, and an effluent flow rate of 180,636 $m^3$/day (Table 1). A schematic diagram of the WWTP treatment process is provided in Figure 1. Previous studies reported that the Ilsanseo-Gu WWTP removes approximately 97.8% of MPs, with a detection of 595.4 MPs particles/L in the influent and 13.1 MPs/L in the final effluent [8]. The Ilsan WWTP operates in multiple phases, with primary settling occurring in the initial treatment phase, followed by a secondary treatment phase that relies on the presence of specific beneficial microorganisms within the biofilms of the PS MPs. In the subsequent tertiary treatment phase, which includes disinfection processes, there is a noticeable trend toward reducing bacterial cell numbers. Therefore, each WWTP treatment stage supports different microbial communities, significantly affecting the composition of the microorganisms in the biofilm.

**Table 1.** Information on the influent and effluent flow rates of the Ilsan WWTP during the incubation period.

| Date | Influent Flow Rate ($m^3$/d) | Effluent Flow Rate ($m^3$/d) |
|---|---|---|
| 21 February | 190,483 | 180,619 |
| 22 February | 198,156 | 189,031 |
| 23 February | 188,177 | 175,208 |
| 24 February | 173,533 | 164,001 |
| 25 February | 183,788 | 188,776 |
| 26 February | 184,733 | 184,461 |
| 27 February | 176,604 | 171,453 |
| 28 February | 185,600 | 174,135 |
| 1 March | 309,517 | 262,622 |
| 2 March | 183,798 | 179,252 |
| 3 March | 184,328 | 176,680 |
| 4 March | 183,367 | 174,389 |
| 5 March | 175,715 | 167,251 |
| 6 March | 183,289 | 177,079 |
| 7 March | 180,822 | 176,478 |
| 8 March | 188,371 | 181,152 |
| 9 March | 186,326 | 182,325 |
| 10 March | 188,041 | 184,768 |
| 11 March | 171,762 | 170,640 |
| 12 March | 182,149 | 171,640 |
| 13 March | 177,429 | 167,192 |
| 14 March | 187,904 | 174,614 |
| Avg. ± sd. | 189,265 ± 27,519 | 180,636 ± 19,500 |

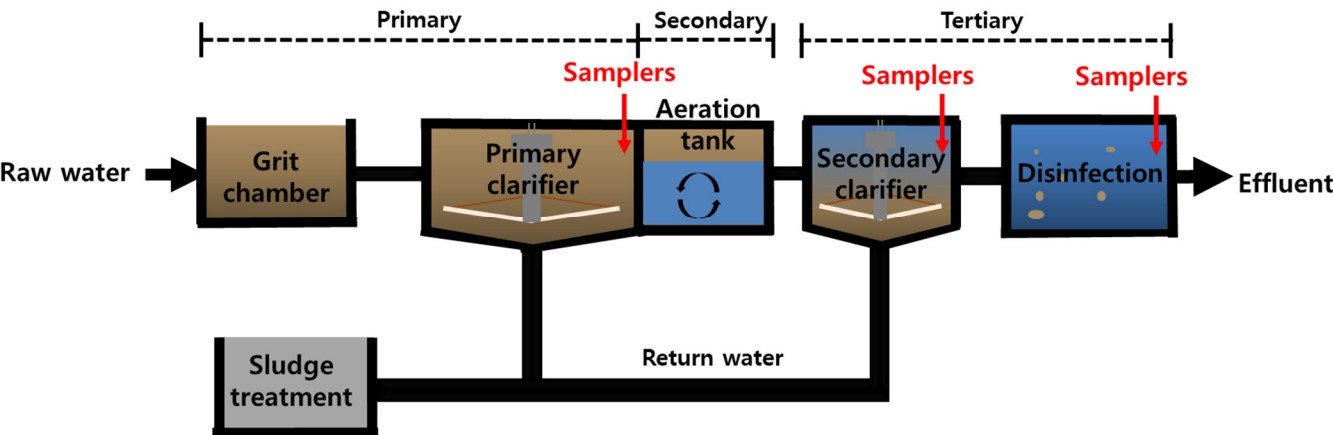

**Figure 1.** Schematic diagram of the WWTP treatment process and the locations of the samplers.

PS MPs with a density of 0.96~1.05 g/cm$^3$, spherical in shape and with a natural color and a size of 2~3 mm, were purchased from GFM in South Korea without any additives. The MPs had dimensions ranging from 2 to 3 mm, and larger sizes compared to microorganisms were deliberately chosen. This choice was made to enable a more precise and focused observation of their interactions. Prior to use, 30 g of PS was washed with deionized (DI) water. Subsequently, the PS MPs were placed in a 7 cm diameter metallic cage with a pore size of 500~600 μm, allowing wastewater and microorganisms to pass through the cage freely while holding the MPs securely. Two of these metallic cages, each containing 30 g of PS MPs, were placed into a single sampler, resulting in each sampler containing 60 g of PS MPs. The samplers were deployed at a depth of 30 cm below the water surface in the WWTP treatment area to simulate the presence of floating MPs in the real world. All samplers were prepared in duplicate. Thus, six samplers, each containing twelve metallic cages filled with PS MPs, were used.

*2.2. Experiment Design: Deployment of MPs in the WWTPs*

Six PS samplers were deployed in total for a duration of 7 days at a depth of 30 cm in the primary treatment stage of the WWTP. After the 7-day deployment period, two out of the six samplers were destructed and analyzed. The remaining four samplers were then transferred to the secondary treatment stage and deployed for an additional week. The rationale behind our decision to use a 7-day timeframe was to create a stable environment conducive to biofilm growth while preventing early colonization, which is typically defined within the first 24 to 48 h. Following 7 days of deployment in the secondary treatment, two samplers containing PS were destructed and stored at −80 °C for subsequent analysis. Finally, the remaining two PS samplers were moved to the tertiary treatment stage and deployed for another 7 days, resulting in a total deployment period of 21 days. Detailed sampler locations are provided in Figure 1.

To obtain a representative sample of the bacterial community in the surrounding water, hereafter referred to as background water, 50 mL of background water was collected from the primary, secondary, and tertiary treatments of the WWTPs. These samples were then compared with the microbial communities present on the MPs.

*2.3. Microbial Analyses*

2.3.1. Morphology Analysis

The morphology of the MPs associated with a biofilm was examined using scanning electron microscopy (SEM). For the SEM analysis, 2~3 MP samples were immersed in 2 mL of 4% paraformaldehyde for 24 h to fix them. Subsequently, the samples were dehydrated using a graded ethanol series in phosphate-buffered saline (PBS), with each concentration (30%, 50%, 75%, and 95%) applied for 10 min, followed by three 15-min incubations in 100% ethanol. After air-drying the samples, they were loaded onto specimen stubs, coated with gold using an ion sputter (E1045, Hitachi, Tokyo, Japan), and examined in secondary electron (SE) mode at 5 kV with the desired magnification.

The cells within the biofilms grown on the surfaces of the MPs were visualized using the fluorescent dye 4′6-Diamidino-2-Phenylindole (DAPI). To prepare the samples, the MPs with a biofilm were fixed in the dark at 4 °C for 30 min. After staining, the MPs were filtered through a 0.2 μm pore size black polycarbonate membrane (ADVANTEC, Whatman) and plated onto a glass microscope slide. The cells stained with DAPI were then observed under blue fluorescence (excitation wavelength of 355~405 nm, emission wavelength of 420~480 nm) with a 30 s exposure time to prevent color fading.

2.3.2. DNA Extraction and High-Throughput Sequencing

DNA extraction was carried out for the background water samples and for the MP biofilm samples which were sequentially deployed from the primary, secondary, and tertiary treatments. For the background water samples, particles and cells in the specified volume (50 mL) of each duplicate sample were collected via centrifugation at 13,000 rpm for 10 min. Regarding the MP biofilm samples, 0.5 g of a PS MP sample was utilized for DNA extraction. A FastDNA Spin kit (MP Biomedical, Santa Ana, CA, USA) was used for DNA extraction, according to the manufacturer's instructions. The extracted DNA was stored at −80 °C for a subsequent molecular analysis.

To target the bacterial 16S rRNA gene, the V3-V4 region was amplified using specific primers, namely 341F (5′-CCTACGGGNGGCWGCAG-3′) and 805R (5′-GGACTA CHVGGGTATCTAATCC-3′). Overhang barcode sequences were attached to these primers to allow for multiplexing and sample identification. The extracted DNA served as the template in the PCR amplification reaction. The PCR products underwent amplicon library preparation, following the Illumina 16S Metagenomic Sequencing Library Preparation Part #15,044,223 Rev. B protocol.

2.3.3. Bacterial Community Analysis

To examine the microbial community structure, the Illumina MiSeq platform (Illumina, San Diego, CA, USA) at Macrogen Inc. (Seoul, Republic of Korea) was employed. The amplicon library preparation followed the protocol of the Illumina 16S Metagenomic Sequencing Library preparation Part #15,044,223 Rev.B. The sequencing data obtained from the samples were analyzed using the Quantitative Insights Into the Microbial Ecology Program (QIIME 2021.4) [27,28].

The sequence data underwent quality filtering, and operational taxonomic units (OTUs) were selected at a 97% identity threshold using relevant plugins, which were consistent with previous research protocols [29,30]. The detailed parameters used in these processes were in accordance with the previous study. For assigning the phylogenetic position to each OTU's representative sequence, the scikit-learn multinomial native Bayesian classifier (ver. 0.24.1), trained with the SILVA database (release ver. 132), was utilized, following the methodology described by Bokulich et al. in 2018 [31–34]. The sequence data used in this study are available from the NCBI Sequence Read Archive (SRA) database under the accession number PRJNA996998.

Statistical analyses were conducted to analyze the feature table of the obtained OTUs. Shannon's diversity and the species richness of the microbial community in each sample were calculated. The Bray–Curtis distance was computed to measure the dissimilarity between samples, and a non-metric dimensional scaling (nMDS) plot was generated for visualization purposes, using a try value of 100. The significance of the differences between groups was assessed using the analysis of similarity (ANOSIM) with 999 permutations. These statistical analyses were performed using the vegan R Package (Version 2.5-7) (https://github.com/vegandevs/vegan, accessed on 28 November 2020) [35–37]. By employing these sequencing and statistical analysis approaches, this study aimed to gain insights into the microbial community structure and diversity associated with MPs in the WWTPs.

## 3. Results and Discussion

### 3.1. The Morphology of the Biofilms on the PS

A SEM analysis was conducted to investigate the morphology of the microorganisms attached on the surface of PS deployed from the primary to tertiary treatments (Figure 2c–e). The results clearly demonstrated the presence of microorganisms on the surface of the PS when deployed in the WWTPs in contrast to the bare MPs, which exhibited only a smooth surface (Figure 2a,b). Upon examining the surface of the PS, we observed heterogeneous assemblages of microorganisms, including both spherical and rod-shaped forms.

Initially, we hypothesized that there would be a distinct difference in the morphology of the microorganisms on the surface of the PS depending on the stages of the WWTP due to the different surrounding water conditions. However, it is difficult to observe the differences in the morphology of the microorganisms among the different treatment stages using the SEM results. However, the SEM data confirmed that the plastisphere was well-structured on the PS deployed in the primary, secondary and tertiary treatments of the WWTP.

### 3.2. Bacterial Community Diversity and Clustering

Bacterial diversity was estimated using $\alpha$-components, including the Shannon index (diversity) and the Chao1 Index (richness), as shown in Table 1. In the background water samples without biofilms, the $\alpha$-diversity of the primary treatment exhibited higher values (4.19 $\pm$ 0.20 for Shannon's Index and 340 $\pm$ 103 for richness) compared to the secondary and tertiary treatments. In the secondary treatment, specific beneficial bacterial strains became predominant, contributing to the effectiveness of biological treatment. In the tertiary treatment, which includes disinfection, the goal is to reduce both the abundance and diversity of microorganisms to prove water clarity. As a result, the inclusion of additional wastewater treatment phases led to a decrease in the diversity of the microorganisms in the water. Specifically, in the secondary treatment, specific beneficial bacteria became

predominant, contributing to the effectiveness of the biological treatment, and the tertiary treatment, which included disinfection, resulted in reductions in both the abundance and diversity of the microorganisms to prove the clarity of the water. Similarly, the biofilm samples found on the PS in the primary treatment displayed higher $\alpha$-diversity indexes than those in the secondary and tertiary treatments.

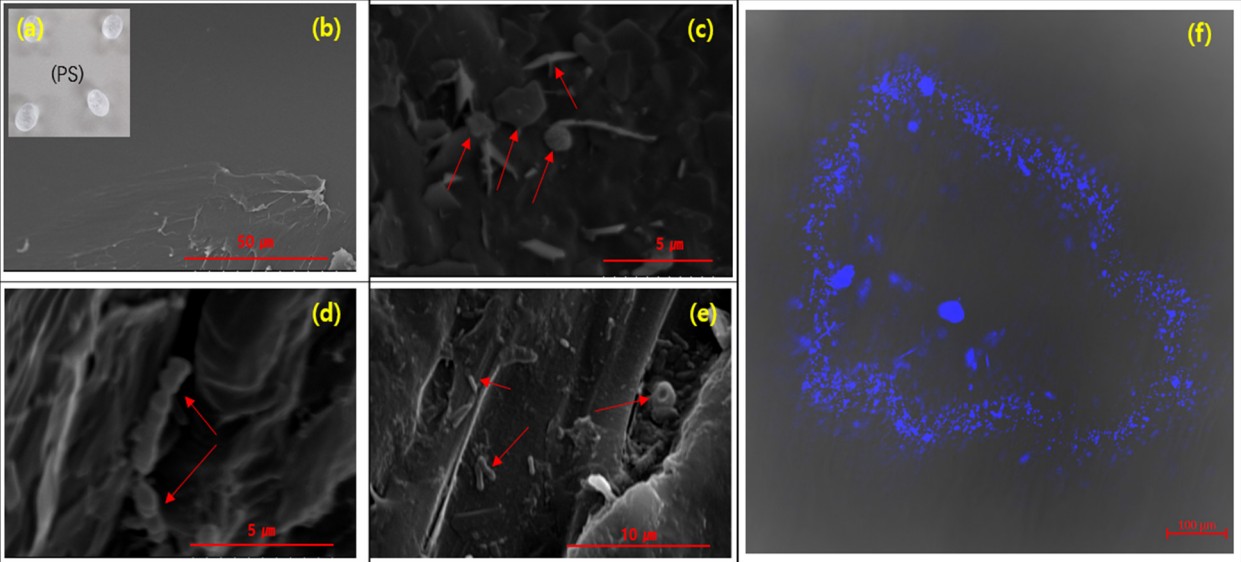

**Figure 2.** SEM images depicting the progression of a microbial biofilm on the PS: (**a**) the original PS, (**b**) PS without a biofilm, (**c**) PS deployed in the primary treatment, (**d**) PS transferred to the secondary treatment, (**e**) PS transferred to the tertiary treatment, and (**f**) fluorescence image of cells attached to the surface of the PS.

When comparing $\alpha$-diversity between the background water samples (without MPs) and the biofilm on the PS, both Shannon's index and richness show an increase of more than ~1 in the latter. This means that the free-floating microbial community in the background water is less diverse than the biofilm development on the PS. However, this result is controversial, as some studies have indicated a decrease in microbial diversity with biofilm development [38,39], while others reported increased diversity in the presence of PS [40–42]. This discrepancy could be attributed to the "umbrella effect" of MPs on the survival and growth of microorganisms in wastewater, but their effects seem to be influenced by the chemical and physical properties of MPs, as well as environmental conditions [43–45]. It is plausible that the formation of an extracellular polymeric substance (EPS) plays a crucial role in the formation, stability, and functioning of microcolonies on the PS. Therefore, in this study, the EPS formed on the PS likely attracts diverse microorganisms from the surrounding water, leading to their accumulation and growth within the biofilm [46–48].

A dendrogram with hierarchical grouping (Figure 3) clearly demonstrates a distinct categorization of bacterial communities between the background water and the biofilms on the PS MP samples. Among the background waters, the samples collected from the secondary and tertiary treatments showed greater similarity compared to the primary treatment. This suggests that while there was a high abundance of diverse, free-floating microorganisms in the primary treatment, their abundance was significantly reduced when they underwent the secondary treatment; this is also supported by the $\alpha$-diversity data presented in Table 2.

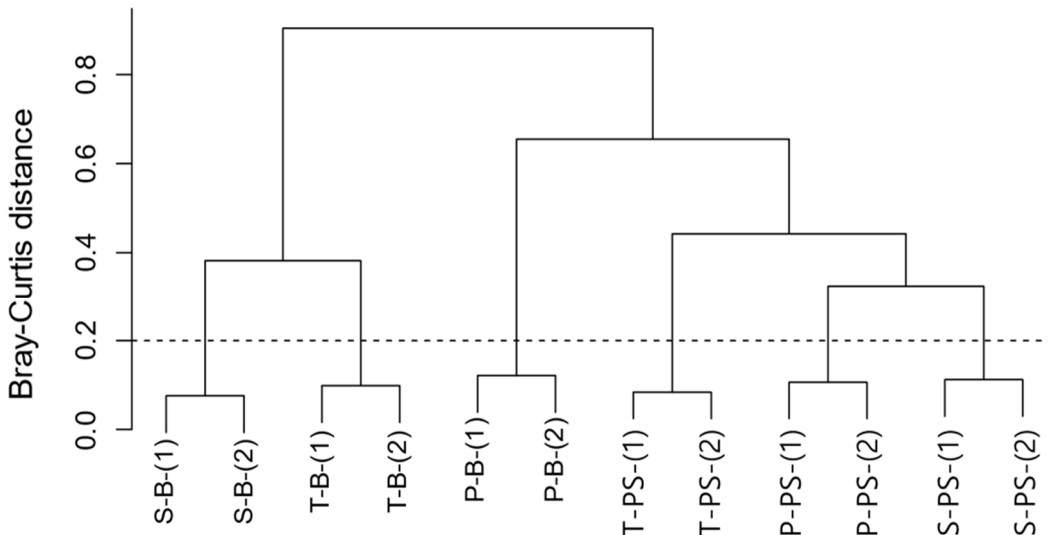

**Figure 3.** Dendrogram of bacterial hierarchical clustering among the background waters and PS biofilm.

**Table 2.** The α-diversity of the microbial communities in the background water and on the PS samples.

| Sample ID | Description | Shannon's Index | Richness |
|---|---|---|---|
| P-B | Background water of primary treatment | 4.19 ± 0.20 | 340 ± 103 |
| S-B | Background water of secondary treatment | 3.10 ± 0.20 | 187 ± 17 |
| T-B | Background water of tertiary treatment | 3.55 ± 0.06 | 275 ± 60 |
| P-PS | PS in primary treatment | 5.06 ± 0.005 | 447 ± 13 |
| S-PS | PS in secondary treatment | 4.97 ± 0.03 | 444 ± 29 |
| T-PS | PS in tertiary treatment | 4.97 ± 0.01 | 420 ± 18 |

However, unlike the background water samples, the biofilm PS samples exhibit a closer distance between the PS deployed in the primary treatment and those subsequently deployed in the secondary treatment. On the other hand, the PS sequentially deployed in the tertiary treatments show distinct patterns compared to the others. Based on this observation, it is hypothesized that initial microbial colonization on the surface of the PS in the primary treatment may induce the attachment of bacteria from the secondary clarifier, persisting through the biological process of the secondary treatment. These attached bacteria may then be detached during their passage through the tertiary treatment, leading to a distinct pattern of bacterial clustering in the biofilm. Thus, these results indicate that the biofilm on the PS is beneficial in resisting the biological treatment process of the secondary treatment compared to the freely floating bacteria in the surrounding waters. However, it is sensitive to being affected by the tertiary chemical treatment.

In the process of the initial formation of a biofilm and its development, environmental conditions such as temperature, pH, oxygen levels, and components play critical roles in determining the biofilm's structure and composition [47,49]. Specifically, *P. aeruginosa*, the most common bacteria known for forming biofilms, exhibits the formation of rod-shaped cells under oxygen-rich conditions, whereas elongated, filamentous cells are formed under oxygen-deprived conditions. Consequently, the structure and characteristics of the biofilm formed are profoundly influenced by these environmental factors [47,49]. This study's results suggest that the microbial community within the biofilm on the PS surfaces varies according to the specific environmental conditions applied in each treatment. This underscores the pivotal role of biofilms on PS as carriers of microbes in the WWTP process.

### 3.3. Bacterial Community Composition and Structure

In both the background waters and PS samples, the bacterial phyla were categorized into five major phyla representing an average abundance of 5% peak abundance in one sample, with additional unclassified bacteria and a minor phylum (Figure 4). At the phylum level, the bacterial communities of all sample types, including background waters and PS, were predominantly composed of Gammaproteobacteria and Bacteroidota. Among the background water samples, the phylum compositions in the secondary and tertiary treatment background waters are more similar when compared to the primary treatment background water. Specifically, in the primary treatment background water, Gammaproteobacteria (58.2 ± 4.4%) dominated the bacterial community, but their relative abundance was reduced in the secondary and tertiary treatments. In contrast, the relative abundances of Patescibacteria and Actinobacteria increased in the secondary treatment background water (17.5 ± 0.2% for Patescibacteria and 11.2 ± 2.8% for Actinobacteria) and tertiary treatment background water (11.9 ± 0.2% for Patescibacteria and 7.8 ± 1.1% for Actinobacteria). The observation that the abundance of Patescibacteria increased in the secondary treatment background water is plausible, considering that Patescibacteria are frequently detected, especially in activated sludges [50–52]. This increase in the abundance of Patescibacteria in the secondary treatment may be attributed to the introduction of return sludge containing a higher concentration of Patescibacteria. In a comparison of the relative abundances of bacterial communities between the background waters and the biofilm on the PS, it was observed that the abundance of Gammaproteobacteria decreased while the abundance of Bacteroidota increased in the PS samples when they were deployed in all WWTP treatments. Gammaproteobacteria are well known as primary colonizers, and Bacteroidota act as secondary colonizers in the formation of biofilms on MPs deployed in various environments such as WWTPs, riverine fresh water, and marine ecosystems [10,38,53–55]. Based on this, a hypothesis can be formulated that the reduced Gammaproteobacteria may function as pioneers in the initial stages of biofilm formation on the PS, and they may have been consumed or outcompeted by Bacteroidota during the later stages. An interesting finding is that the proportion of Firmicutes in the PS was notably increased compared to the background water samples, particularly in the primary treatment. Firmicutes are known to be chemically degrading bacteria and are strongly resistant to extreme environments [56,57]. Therefore, it is plausible that Firmicutes preferentially attach to the PS due to their capability to degrade organic functional groups in the PS, as well as other bacteria and nutrients adsorbed on the biofilm on the PS. This may contribute to their higher abundance in the biofilm formed on the PS compared to the background waters.

To track the evolution of bacterial communities in the biofilm of PS undergoing treatments, from the primary to tertiary stages, it was observed that there was little difference in the proportion of Bacteroidota between the primary and secondary treatments, with values of 41.5 ± 0.3% and 40.0 ± 2.7%, respectively. A noticeable decrease in abundance was observed in the tertiary treatment (25.8 ± 0.4%). Additionally, the abundance of Firmicutes continuously decreased as the PS underwent the WWTP process from the primary to tertiary treatments, showing 18.2 ± 0.6% for the primary treatment, 11.2 ± 0.1 for the secondary treatment, and 4.7 ± 0.1% for the tertiary treatment. This suggests that Firmicutes, which attach to the biofilm by consuming organic/inorganic materials present in the biofilm, may be detached due to the depletion of these materials in the biofilm as the WWTP treatments are added, either indirectly and/or by the action of disinfection treatments that occur in the tertiary stage. These observations indicate that the composition of the bacterial communities in the biofilm of PS undergoes dynamic changes throughout the WWTP process. The decreases in the abundances of Bacteroidota and Firmicutes in the tertiary treatment suggests that certain bacterial groups may be more sensitive to the changing conditions and treatments in the later stage of the WWTP process.

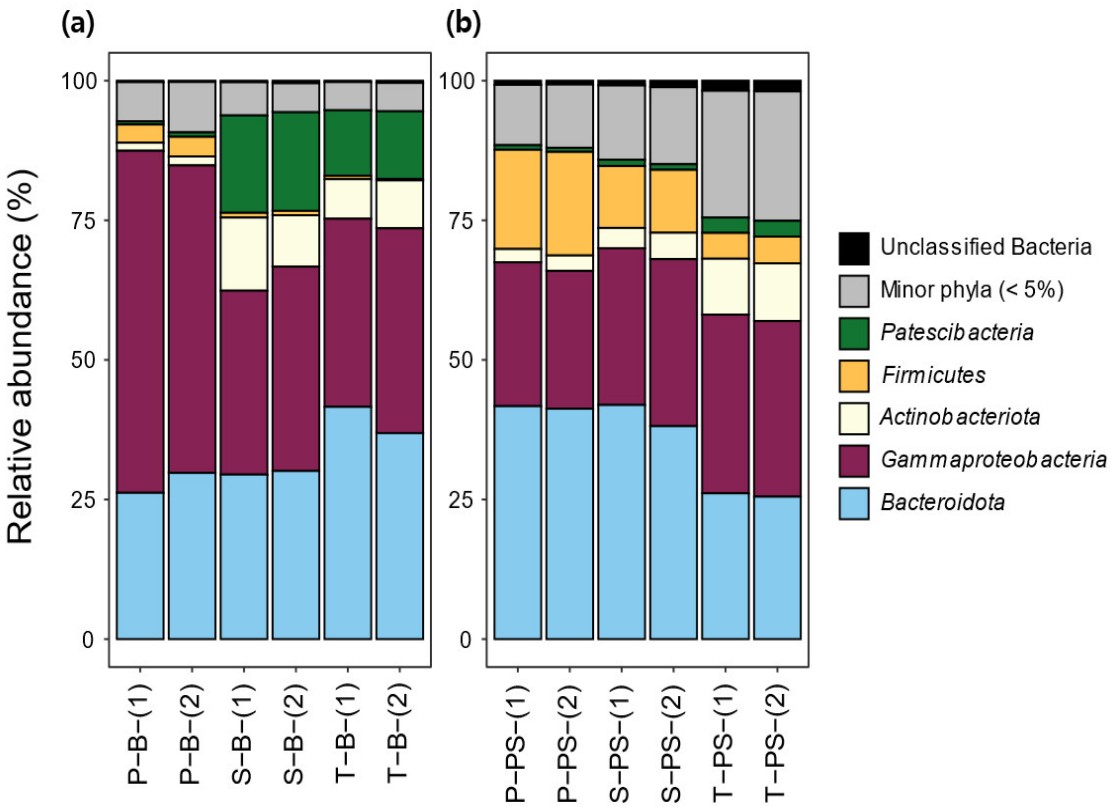

**Figure 4.** Phylum-level bacterial communities in the background water (left; (**a**)) and the surface of PS throughout WWTP stages (right; (**b**)). P—primary treatment, S—secondary treatment; T—tertiary treatment. Hight-throughput sequencing was conducted in duplicate (-(1) and -(2)) for each sample. Bacterial phyla which accounted for less than 5% across the samples were summed as "Minor phyla (<5%)".

### 3.4. In-Depth Microbiome at the Genus Level

The bacterial communities in both the background waters and biofilms on the PS were analyzed at the genus level to gain comprehensive insights into their compositional structures and functional attributes (Figure 5). Comparing the relative abundances of genera between background waters and the biofilms on the PS revealed a more diverse distribution of genera on the PS, with certain specific genera showing higher relative abundances (20~25%) in the background waters, while all the genera found on the PS had a relative abundance smaller than 10%.

In the background waters, the relative abundances and compositions of genera in the secondary and tertiary treatments were similar, whereas the primary treatment background water differed significantly. The primary treatment background water had a more diverse distribution of genera, while both the secondary and tertiary treatment background waters exhibited a more focused distribution, indicating that diverse genera in the primary treatment were reduced or eliminated by going through the secondary and tertiary treatments. Notably, Polarmonas dominated the primary treatment background water but was absent in the secondary and tertiary treatments. Conversely, Flavobacterium and Saccharimonadales increased in both the secondary and tertiary treatments. Additionally, Polynucleobacter and Fluviicola, which were not present in the primary treatment background waters, were found in the secondary and in the tertiary treatments, respectively, which may be hypothesized to be due to the introduction of return sludge.

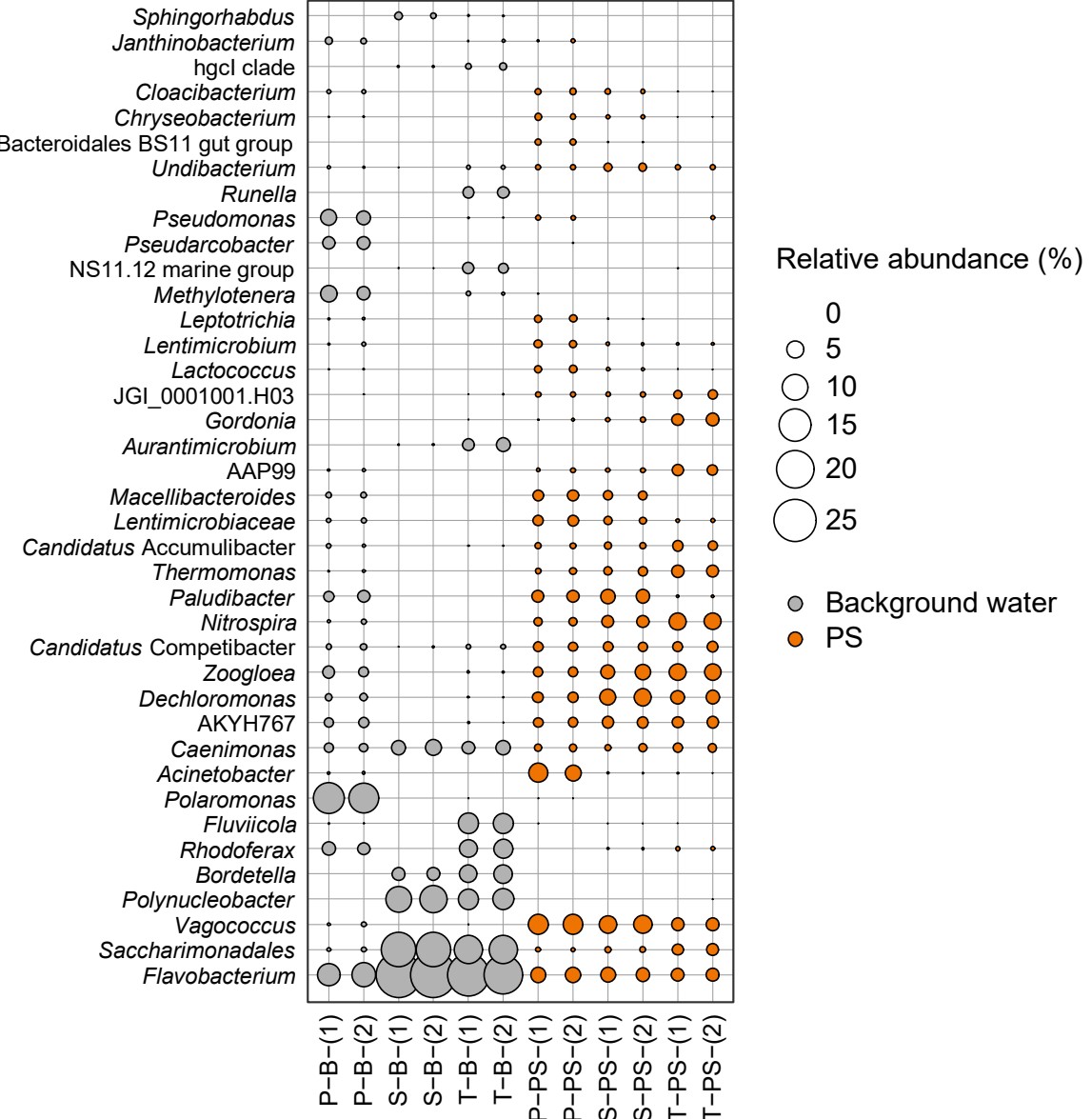

**Figure 5.** Relative abundances (%) of genera observed in the background water samples from primary, secondary, and tertiary treatments (represented in gray) and on the PS surfaces deployed in the primary, sequentially secondary, and tertiary treatments (represented in orange). The size of each circle corresponds to the % of relative abundance.

In the biofilms on the PS MP samples, similar to the background water samples, the pattern of genera found in the secondary treatment was also present in the tertiary treatment, differing from that of the primary treatment. While a significant difference at the genus level was observed between the background water of the primary treatment and the background water of the secondary and tertiary treatments, not much difference was observed in the PS samples among the treatment stages. This indicates that once they were attached to the surfaces of the PS MPs in the primary treatment, their relative abundance was either increased or decreased but not completely altered.

For the PS deployed in the primary treatment, Psedomonas, Pseudarcobacter, Methylotenera, Polaromonas, and Rhodoferax almost disappeared or significantly diminished in the biofilm on the PS compared the primary treatment background water, suggesting they do not prefer to attach to the PS surface PS. Conversely, the relative abundances of Vagococcus and Acinetobacter increased on the PS samples deployed in the primary treat-

ment, indicating their preference for attachment to PS. Acinetobacter, known for causing human infectious diseases such as nosocomial infections, was also reported to be more abundant on effluent MPs, suggesting that bacterial assemblages colonizing MPs can influence their fates within WWTPs [20,58]. In addition, Kelly et al. (2021) demonstrated a high abundance of *Psedomonas* on the surfaces of MPs present in an effluent [20]. Additionally, *Pseudarcobacter* constituted 29.5% of the microbial communities on the MPs [59], which supports the selective attachment of this species to the surfaces of MPs.

Contrary to the initial hypothesis, approximately 10 out of 25 genera found in the primary treatment showed consistent or increased relative abundances as the PS traversed the WWTP treatments. These genera, including JGI_0001001.H03, Gordonia, AAP99, Candidatus Accumulibacter, Thermomonas, Nitrospira, Zoogloea, Dechloromonas, and Saccharimonadales, may act as pioneers, attracting and facilitating the accumulation and growth of the certain genera on the PS with resistance to disinfection treatment, leading to an increased abundance in the biofilm. Thus, taken together, it can be inferred that the microbes present in the background waters selectively colonized the surfaces of the PS MPs. The microbial communities attached to the PS MPs in the primary background acted as pioneers, facilitating the accumulation and growth of other subsequent microbes, leading to the formation of succession biofilms. Additionally, these biofilms formed on the PS may play a role in resisting the disinfection processes in WWTPs, potentially altering their fate as carriers of microbes in the effluents of WWTPs.

Many reports have revealed that the surfaces of MPs in WWTPs can serve as a breeding grounds for pathogens, as well as antibiotic resistance bacteria (ARBs) and antibiotic resistance genes (ARGs) [60–62]. For example, Perveen et al. (2023) [11] reported that Pseudomonas, Aeromonas, and Bacillus among ARBs and intl1 existed with high relative abundances on the surface of polystyrene (PS) MPs in a WWTP effluent. Another study showed that polyethylene (PE) MPs function as reservoirs of Thermoanaerobacter, Tepidimicrobium, Sporanaerobacter, Lutispora, Caldicoprobacter, and Methanothermobacter, which are known as ARBs, in the activated sludge of WWTPs [63]. Therefore, considering the results of previous studies and these observations in this study highlights the importance of understanding the dynamics of microbial communities on MPs within WWTPs and their potential implications for environmental and public health.

## 4. Conclusions

This study investigated how microbial communities on PS change as they traverse through different stages of WWTPs, including primary, secondary, and tertiary treatments. During the primary treatment, microbial colonization on PS sets the stage for subsequent colonization, and biofilms offer favorable environments for robust microorganisms. Thus, certain microbes from background waters exhibited a preference for attaching to the surfaces of PS MPs, with the initial microbial communities in the primary treatment acting as pioneers. This transition led to alterations in the bacterial communities in the biofilm, with enhanced resistance to disinfection processes, maintaining or even increasing their abundance as they progressed through the WWTP treatments.

The presence of diverse microbial communities on the PS and their resistance to treatment processes raise concerns about MPs as potential carriers of microbial pollutants in WWTP effluents. The persistence of pathogenic bacteria and antibiotic resistance genes on MPs within the plastisphere emphasizes the need for effective strategies to control MPs and associated microbial pollution in WWTPs.

In conclusion, this study enhances our understanding of the formation and dynamics of microbial communities on MPs within WWTPs. This information is beneficial for developing strategies for the removal of MPs and elucidating the fate of MPs and associated microbes within WWTPs. The formation of biofilms on MPs is a critical aspect, emphasizing their role as carriers of diverse microorganisms and their implications for environmental and public health. Further efforts should focus on understanding the factors influencing

biofilm development on MPs and their fate during WWTP processes, contributing to a healthier and more sustainable environment.

**Author Contributions:** J.-K.H. performed the bacterial analysis and wrote the manuscript; H.O. provided constructive discussions regarding this study; T.K.L. reviewed the manuscript and provided comments on the results and discussion; S.K., D.O. and J.A. conducted the literature review that informed the direction of this study; S.P. contributed to the study's conceptualization, reviewed and edited the manuscript, and provided overall supervision for this study. All authors have read and agreed to the published version of the manuscript.

**Funding:** This research was funded by the KICT research program through the Ministry of Science and ICT (project. no. 20230160-001, Research on Next Generation Environmental Technology for Carbon Neutrality) and supported by the National R&D program through the National Research Foundation of Korea (NRF), funded by the Ministry of Science and ICT, Republic of Korea (NRF-2021M3E8A2100648).

**Institutional Review Board Statement:** Not applicable.

**Informed Consent Statement:** Not applicable.

**Data Availability Statement:** The data presented this study are available upon request from the corresponding author.

**Acknowledgments:** The authors are very grateful for the funding (project no. 20230160-001 and NRF-2021M3E8A2100648) provided by the Ministry of Science and ICT.

**Conflicts of Interest:** The authors declare no conflict of interest.

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
