# Peer review of "Tracking the Evolution of Microbial Communities on Microplastics through a Wastewater Treatment Process: Insight into the “Plastisphere”"

_water, doi:10.3390/w15213746_

Round 1
Reviewer 1 Report
The study investigated the microbial composition on PS microplastics surface, and compared with the surrounding wastewater microbiota. They found PS can selectively absorb specific microbiota. This is interesting as the microplastics have been found to be a prevalent carrier of microbiota. The study is useful to show the interaction between microplastics and microbiota. Some concerns are listed as below,
1. Arrows can be deployed to show the existence of bacteria in SEM in figure 2.
2. The quality of metabarcoding sequence should be verified, such as, Did the sequencing depth is high enough to obtain the most reads.
3. It would be better to carry the test to check the existence of biofilm on PS particles, as this is clearly mentioned in the manuscript.
4. As the colonization of microbiota on PS particles is largely influenced by surrounding environment, the pH, temperature, turbidity, and other factors should be presented to better define the environment of PS particles.
5. The authors showed selective attachment of certain species. However, as the limited samples performed in the study, random selection could still be an explanation. Please give more discussion, and also try to integrate previous studies to dig the details.
The English is clear.
Author Response
We appreciate the reviewers’ time to provide us with the comments. We have addressed the specific comments of the reviewer as below and in most cases, have made associated changes to the manuscript and/or SI as detailed below.
Please see the attachment.

Reviewer 2 Report
General
This review aims to shed light on the evolution of microbial communities on MPs within WWTPs and their role as carriers of microbes in effluents: Tracking the Evolution of Microbial Communities on Microplastics through a Wastewater Treatment Process: Insight into the ‘Plastisphere’. The evolution of microplastic spheres in wastewater treatment plants was simulated by analysing the microorganisms on the surface and in the environment of microplastics at different stages. This is a topic of interest to readers in related fields. However, the research paper still needs to be improved in terms of experimental design and discussion of results. In order to better discuss the results, you should improve the clarity of the figures and adjust the context of the article. Therefore, I suggest that the manuscript be revised before publication.
The following are my specific comments
1. The line numbers in the text are suggested to be changed to be consecutive Introduction.
2. Page 1: There is something wrong with the expression of this sentence “As a consequence, MPs are ubiquitous in various environmental compartments, including soils, aquatic environments, wildlife, and the atmosphere [5-8].”What’s mean of “environmental compartments”?
3. Page 1: The sentence of “Recent studies have revealed that MPs serve as novel habitats for diverse microorganisms, including pathogenic bacteria, antibiotic resistance genes (ARGs), and metal-resistant genes (MRGs) [11-14].” is not involved in the following. I suggested removed it.
4. Page 1: The “Thus” in the sentence of "...mimicking specific treatment stages of WWTPs, or deploying them in-situ during only target WWTP stages. Thus, they lack...” is improper.
5. Page 2: Why is the size of microplastics type chosen in the experiment 2-3mm? What is the reason for the test period of 7d?
Materials and Method:
6. Page 2: I suggest that modify the “2.1. Description of the WWTP” into a Table. Moreover, please reply more physical and chemical information about WWTPs which is related to microbial growth.
7. Page 2: Only six samplers are placed in WWTPs? How is the concentration of the sampler set? What is the size of the sampler? What is the basis for choosing the water layer height of 30cm?
8. Page 3: Please explain why you use ethanol to wash the MPs.
Results and Discussion
9. Page 4: The pictures in the manuscript are not clear, please modify the corresponding parts involved in the manuscript.
10. Page 5: As for the part of “Bacterial community diversity and clustering”, I suggest replenish a significant analysis for diversity and richness.
11. Page 5: The sentence of “This could be attributed to the addition of further wastewater treatments, which resulted in a decrease in the diversity of microorganisms in the polluted water.” is indigestible. It is not clear here how the further treatment of wastewater leads to the decline of diversity.
12. Page 6: From what you said above, the influence of richness and diversity comes from many aspects. Why are you sure that EPS is an important indicator leading to the change of microbial richness and diversity in your article?
13. The tense of the discussion part in the manuscript cannot be suppressed, so it is suggested to check the whole manuscript.
14. Why the description in the article falls to the analysis of microorganisms at the level of genus in part 3.4. In fact, the microbial community has been analyzed from part 3.3, and it is suggested to merge 3.3 and 3.4.
15. Although the abstract of the article mentioned the risks of pathogenic bacteria and so on. In fact, there is no discussion about this part in the article. It is suggested to modify it to make the summary conform to the discussion.
16. Some middle-aged people discussed in the article do not explain the reasons for the change of microbial abundance well. It is suggested to illustrate the description of the changing trend in the article.
Conclusion
17. The conclusion is not concise, and it can't accurately tell the main idea of the article. It is suggested to modify it.
Overll, I think that the context of manuscript can be readable, and English Language is good.
Author Response
We sincerely thank you for your valuable time and for sharing your insightful comments. We have carefully revised the manuscript, incorporating the majority of your suggestions. In doing so, we have aimed to enhance the discussion of our results and improve the clarity of our figures and tables. Your feedback has been instrumental in refining our work.
Please see the attachment.
